# A Mutation in Mouse MT-ATP6 Gene Induces Respiration Defects and Opposed Effects on the Cell Tumorigenic Phenotype

**DOI:** 10.3390/ijms24021300

**Published:** 2023-01-09

**Authors:** Raquel Moreno-Loshuertos, Nieves Movilla, Joaquín Marco-Brualla, Ruth Soler-Agesta, Patricia Ferreira, José Antonio Enríquez, Patricio Fernández-Silva

**Affiliations:** 1Department of Biochemistry and Molecular and Cell Biology, University of Zaragoza, 50009 Zaragoza, Spain; 2Institute for Biocomputation and Physics of Complex Systems, University of Zaragoza, 50018 Zaragoza, Spain; 3Department of Mechanical Engineering, Multiscale in Mechanical and Biological Engineering (M2BE), Aragon Institute of Engineering Research (I3A), University of Zaragoza, 50018 Zaragoza, Spain; 4Centro Nacional de Investigaciones Cardiovasculares (CNIC), 28029 Madrid, Spain

**Keywords:** ATP synthase, mitochondria, OXPHOS, disease

## Abstract

As the last step of the OXPHOS system, mitochondrial ATP synthase (or complex V) is responsible for ATP production by using the generated proton gradient, but also has an impact on other important functions linked to this system. Mutations either in complex V structural subunits, especially in mtDNA-encoded *ATP6* gene, or in its assembly factors, are the molecular cause of a wide variety of human diseases, most of them classified as neurodegenerative disorders. The role of ATP synthase alterations in cancer development or metastasis has also been postulated. In this work, we reported the generation and characterization of the first *mt-Atp6* pathological mutation in mouse cells, an m.8414A>G transition that promotes an amino acid change from Asn to Ser at a highly conserved residue of the protein (p.N163S), located near the path followed by protons from the intermembrane space to the mitochondrial matrix. The phenotypic consequences of the p.N163S change reproduce the effects of MT-ATP6 mutations in human diseases, such as dependence on glycolysis, defective OXPHOS activity, ATP synthesis impairment, increased ROS generation or mitochondrial membrane potential alteration. These observations demonstrate that this mutant cell line could be of great interest for the generation of mouse models with the aim of studying human diseases caused by alterations in ATP synthase. On the other hand, mutant cells showed lower migration capacity, higher expression of MHC-I and slightly lower levels of HIF-1α, indicating a possible reduction of their tumorigenic potential. These results could suggest a protective role of ATP synthase inhibition against tumor transformation that could open the door to new therapeutic strategies in those cancer types relying on OXPHOS metabolism.

## 1. Introduction

Although ATP synthesis, using the electrochemical proton gradient generated in the mitochondrial electron transport chain, is the main function of F_1_F_o_ ATP synthase (EC 7.1.2.2 or complex V) [1], this enzyme is also involved in other pathways that have an impact on relevant functions connected to the OXPHOS system [2]. For instance, ATP synthase has an important role in shaping the structure of mitochondrial cristae together with other inner mitochondrial membrane (IMM) proteins, such as OPA1 (optic atrophy 1) and some components of the MICOS (Mitochondrial contact site and cristae organizing system) complex [3,4,5]; it is involved in the permeabilization of the IMM under physiological and pathological conditions [6,7] and participates in the control of intracellular signaling pathways [8], regulating cell death and survival [9,10].

Mitochondrial ATP synthase is a rotary enzyme composed of 17 different protein subunits with two of them, subunits a and A6L, encoded within mtDNA [11]. This large complex is organized in two functional domains: the hydrophobic one (F_o_), embedded in the IMM, and the catalytical domain (F_1_), located in the matrix and responsible for ADP phosphorylation to produce ATP. Both domains are connected by two stalks that couple proton transfer through F_o_ from the intermembrane space (IMS) to the matrix, to ATP synthesis, in F_1_ [1,12].

Due to the crucial role of ATP synthase for maintaining cellular functions and proliferation, mutations in either complex V structural subunits or in its assembly factors are the molecular cause of a wide variety of human diseases, mostly neurodegenerative disorders [13]. The majority of the disorders affecting mitochondrial complex V characterized to date are due to mutations in the mitochondrial gene encoding subunit a *MT-ATP6*. This is a transmembrane protein located in the F_o_ domain, where it participates in two connected functions: first, it takes part of the channel that conducts protons through the IMM to the matrix and second, it interacts with the ring of c-subunits promoting its rotation driven by proton pumping [1]. The resulting mechanical energy, conducted through the stalk, induces conformational changes in F_1_ domain to allow ATP production [1,12,14,15]. Moreover, this subunit has also been suggested to have an important role in complex V dimerization and stabilization. This process takes place via the F_o_ domain, and subunit a constitutes the most important basis for dimerization as it presents a high number of transmembrane helixes [1,16,17]. The most frequent mutations in the *MT-ATP6* gene affect the same amino acid residue of subunit a, p.L156R (m.8993T>G) and p.L156P (m.8993T>C) and have been identified in numerous patients who present the NARP (neuropathy, ataxia, and retinitis pigmentosa) or MILS (maternally inherited Leigh syndrome) syndromes, depending on the mutation load. The m.8993T>G mutation compromises severely ATP production (up to 90%), due probably to a block in proton translocation or to defects in coupling or in assembly of ATP synthase [15,18,19,20,21,22,23,24,25]. In contrast, the second mutation (m.8993T>C) promotes a less severe pathological phenotype with a decrease in ATP production of 70%, due mainly to a reduction of assembly or stability of subunit a [22,26,27,28,29]. Two mutations at amino acid 217, p.L217R (m.9176T>G) and p.L217P (m.9176T>C) are also frequent and promote similar diseases to the ones associated to position 156 changes. Whereas p.L217R is extremely severe as it blocks the assembly of subunit a, the second one affects ATP synthase function without compromising the assembly [30]. Finally, the m.9185T>C (p.L220P) mutation has been found in patients with MILS and other neurological disorders [31,32] and promotes a drop of 50–90% in ATP production, due to changes in the enzyme activity. Moreover, other less frequent mutations have been described in MT-ATP6 up to date, promoting a wide variety of clinical symptoms including Leber hereditary optic neuropathy (LHON) (for a review see [30]).

Recently, ATP synthase functions have been associated in the literature with different properties of cancer cells such as proliferation behavior, and cell death evasion (reviewed in [33]). Several mutations within the MT-ATP6 gene have been found in different human carcinomas, promoting tumor growth by reducing cell death [34,35]. In addition, changes in H^+^-ATP synthase subunit expression in cancer cells have been reported. For example, a downregulation of the catalytic subunit β-F_1_-ATPase expression has been found in the most prevalent human carcinomas when compared to normal tissues [36,37]. Furthermore, several tumor types such as colon, lung, breast and ovarian carcinomas, among others [38], show an upregulation of the expression of the ATPase inhibitor IF1 in order to favor cell proliferation by preserving the ATP pool [38,39]. For these reasons, mitochondrial ATP synthase has been proposed as a possible therapeutic target against cancer progression and metastasis [40,41].

Here, we reported the generation and characterization of the first *mt-Atp6* pathological mutation in mouse cells, an m.8414A>G transition that promotes an amino acid change from Asn to Ser. This change affects a highly conserved residue of the protein (p.N163S), located in the aH5 helix near the path of protons from the IMS to the mitochondrial matrix and in a position laying between two pathological mutations previously reported in humans [42,43]. The phenotypic consequences of the p.N163S substitution reproduce many of the effects of MT-ATP6 mutations in human diseases, such as increased dependence on glycolysis, defective OXPHOS activity, ATP synthesis impairment, increased reactive oxygen species (ROS) generation, mitochondrial membrane potential alteration and F_1_-subcomplex accumulation. These observations demonstrate that this mutant cell line could be a useful tool to generate mouse models to study human diseases caused by alterations in ATP synthase. Modelling of the mutation effects on subunit a structure and interactions suggests an alteration in proton flow and in F_o_ rotation that could explain the ATP synthesis defect. On the other hand, the analysis of mutant cells revealed a lower migration capacity, higher expression of major histocompatibility complex I (MHC-I) and lower levels of hypoxia inducible factor 1α (HIF-1α), suggesting a reduced tumorigenic potential. These preliminary results might indicate a protective role of ATP synthase inhibition against tumor transformation that could be explored to find new therapeutic strategies in some types of cancer with an OXPHOS-dependent metabolism.

## 2. Results

### 2.1. Isolation of a Mouse Cell Line with an mtDNA Driven Mutation in ATP Synthase

A mutagenesis and selection strategy developed to generate mtDNA mutations [44] was applied to obtain new mouse cell lines that could represent models for human mtDNA-associated mutations. Briefly, a control cell line, TmBalb/cJ, obtained by transference of mitochondria from Balb/cJ mouse platelets to mt-DNA depleted cells (ρ^0^L929^neo^) as previously described [45], was treated with trimethyl-psoralen (TMP) in order to randomly induce mutations in mtDNA [44,46]. After treatment with ethidium bromide (EthBr) to reduce the mtDNA copy number, subcloning and recovery to allow mtDNA repopulation, a selection in a galactose-containing medium was applied and a potential OXPHOS defective clone (BcTMPII.1) was selected for further analysis. Full mtDNA sequencing of the potentially mutant clone showed the presence of a single mutation, an A to G transition at position 8414 (Figure 1A) within the *mt-Atp6* gene in homoplasmic form, as confirmed by RFLP analysis (Figure 1B). This mutation promotes an amino acid change from asparagine to serine at position 163, which is a highly conserved exposed and functional residue located in the aH5 helix of the F_o_ ATPase a subunit (MT-ATP6) (Figure 1C), forming part of a positively charged region around the highly conserved Arg159, essential for the transfer of protons from the IMS to the mitochondrial matrix [30] (Figure 1D). Multiple protein sequence alignment against 100 non-redundant UniProtKB/SwissProt sequences, including animals (vertebrates and invertebrates), plants, fungi and bacteria, showed 100% conservation in this position among the 99 species, presenting more than 165 amino acids in subunit a. As shown in Appendix A, both mt-Atp6 gene and protein sequences present variable identity scores among different organisms. In addition, different amino acid substitutions in MT-ATP6 are responsible for human pathologies such as NARP/MILS, LHON or Leigh diseases (Figure 1E) among others, several of them being located adjacent or very close to the equivalent of mouse 163 position mutated in the clone BcTMPII.1 and affecting aH5 helix.

Although the mutagenesis and selection methods applied mainly lead to mtDNA mutations, they could also affect nuclear genes. To exclude the effects of potential nuclear gene mutations, we performed mitochondrial transfer from both control and BcTMPII.1 cell lines into a receptor cell line lacking mtDNA (ρ^0^L929^hygro^), thus generating transmitochondrial clones harboring the same nuclear background, named TmControl and TmA6^MUT^, respectively, and those were used for further analysis.

### 2.2. Cells Carrying the m.8414A>G mtDNA Variant Are Defective in OXPHOS Performance and Dependent on Glycolysis

Cells with mtDNA mutations usually present variable difficulties to grow in a culture medium where glucose is replaced by galactose as energy source, depending on the degree of the OXPHOS impairment. As shown in Figure 2A, TmA6^MUT^ cells had a parallel growth curve to control cells in glucose medium but were unable to grow in the presence of galactose, whereas the control cells showed similar doubling times in both glucose and galactose media. This result strongly supports the idea that mutant cells have a severe defect in OXPHOS and rely mainly on glycolysis for energy supply. The observation of a higher glucose consumption and lactate production in the glucose-containing culture medium by TmA6^MUT^ cells after an identical time in culture and equivalent cell density (Figure 2B; also confirmed by the faster acidification of the medium (Appendix A)), reinforces this interpretation.

Since ATP synthase activity is coupled to mitochondrial respiration, we measured oxygen consumption rate in intact control and mutant cells. As shown in Figure 2C, mutant cells present significantly impaired coupled mitochondrial respiration compared to control cells (left panel). Uncoupled respiration (Figure 2C, right panel), which is an index of maximal respiratory capacity independent of ATPase activity [47], confirmed this impairment. To determine which complexes were limiting respiration capacity in mutant cells, we performed polarographic measurements of respiratory chain complexes in digitonin-permeabilized cells. As illustrated in Figure 2D, the reduction in respiratory capacity in the mutant line is maintained at any entry point of electrons, suggesting that the whole electron transport chain is affected. However, the analysis of in-gel activity assay (IGA) for complex IV, that can reflect the isolated activity of this complex, did not reveal significant differences between control and mutant cells (Figure 2E).

The previous results strongly pointed to a higher dependence of the mutant cells on glycolysis due to damage in the mitochondrial function. To further test this possibility, we compared the sensitivity of control and mutant cells to drugs that inhibit glycolysis, testing different concentrations and determining their effect on cell viability. Thus, we estimated the inhibitory concentration 50 (IC_50_) for sodium iodoacetate (IA), which irreversibly inhibits the glycolytic enzyme glyceraldehyde-3-posphate dehydrogenase (GAPDH) [48], and sodium dichloroacetate (DCA), which inhibits pyruvate dehydrogenase kinase, promoting pyruvate conversion into acetyl-CoA. As an additional control, we evaluated the cytotoxic effect of both inhibitors in ρ^0^ cells, that are completely dependent on glycolysis. As shown in Figure 2F, TmA6^MUT^ cells are significantly more sensitive to IA than control cells confirming their higher dependence on glucose metabolism and they are also more sensitive to DCA, confirming that their damaged OXPHOS system cannot fully compensate the reduced glycolytic route.

### 2.3. Mutant Cells Show Impaired ATP Synthesis and Differences in Mitochondrial Membrane Potential

In order to investigate if the mutation in *mt-Atp6* gene promotes impairment in ATP synthase activity, we evaluated the mitochondrial ATP synthesis rate of mutant compared to control cells. ATP synthesis was assayed in digitonin-permeabilized TmControl and TmA6^MUT^ cells using pyruvate and malate as substrates. As it can be seen in Figure 3A and Appendix A, ATP synthesis is severely hampered in TmA6^MUT^ cells, being produced at a rate of 11% relative to control cells (Figure 3A). The treatment with the specific inhibitor of the ATP synthase F_o_ fraction oligomycin, revealed that mutant cells were significantly more sensitive (lower IC_50_) than control cells (Figure 3B). This result could be explained by the fact that their OXPHOS system is already partially damaged and hence its further inhibition causes that a threshold affecting viability (probably through essential routes connected with this activity such as nucleotide synthesis or redox balance) is reached at a lower oligomycin concentration than in cells with a stronger OXPHOS [49].

Mitochondrial membrane potential (ΔΨm) alterations can be caused by the accumulation of protons in the IMS or by an abnormal flux to the matrix coupled to ATP synthesis. Since TmA6^MUT^ cells maintained a substantial respiration capacity (although reduced when compared with control cells) and ATP synthesis was severely hampered, one could expect alterations in their ΔΨm. To test this possibility, we measured ΔΨm by cytometry, using DiOC_6_ staining. As shown in Figure 3C, we observed a slight but not significant increase of ΔΨm in mutant cells. To further analyze this aspect, we cultivated both cell lines in the presence of different concentrations of DCA to force OXPHOS and we measured ΔΨm again. As shown in Figure 3D, although the mean fluorescence intensity (MFI) in mutants seems to decrease after treatment with 25 mM DCA, two different peaks can be distinguished: one of them matches with the size and complexity of normal-sized healthy cells, while the other one corresponds with shrunk cells, which are thus significantly affected by DCA treatment and probably dying. Considering exclusively the peak of healthy cells, the ΔΨm increases significantly when compared to that of mutant untreated cells or control cells grown in the presence of 25mM DCA (MFI: 513 vs. 345 and 484 respectively, Figure 3E). Therefore, this hyperpolarization observed when OXPHOS is forced and glycolysis inhibited, indicates an abnormal proton flux through the ATP F_o_ portion in the mutant cells. The measurement of ΔΨm by cytometry with a different dye, TMRE, and in the presence of an uncoupler, FCCP, confirmed the validity of the previous data and the existence of differences between both cell lines (Appendix A).

### 2.4. m.8414A>G Mutation Induces Differences in Mitochondrial Mass and ROS Production

Due to defects in F_1_F_o_ ATP synthase function, human cybrids harboring mt-ATP6 mutations have been reported to produce higher basal levels of mitochondrial ROS (mtROS) than their control counterparts [1,15,50,51]. For this reason, we sought to measure mitochondrial superoxide generation in our cell lines using MitoSOX^TM^. As shown in Figure 4A, mutant cells generate around 30% more mitochondrial superoxide than control cells. To determine whether the increase in ROS could trigger a rise in the number of mitochondria or on mtDNA levels as an adaptive response, as has been reported in some cases [52,53], we evaluated mitochondrial mass. However, as shown in Figure 4B, both mtDNA copy number, measured by q-PCR, and mitochondrial mass, analyzed by MitoTracker Green staining followed by flow cytometry, indicated a significant reduction in mitochondrial mass in mutant cells. When normalizing ROS levels to mitochondrial mass, the differences between control and A6 mutant cells become much higher (Figure 4C). If the respiration values shown in Figure 2C,D are also normalized by mitochondrial content to better evaluate the respiratory chain efficiency, a significant reduction is still detected in mutant cells with respect to control cells in basal (coupled) respiration (Figure 4D). However, normalization of the respiration observed in permeabilized cells with complex-specific substrates, shows differences between mutant and control cells that become non-significant (Figure 4E), suggesting that the main defect induced by the mutation involves primary CV and not the respiratory chain efficiency, although a general reduction in respiration activity is developed in mutant cells.

### 2.5. m.8414A>G Mutation Does Not Affect ATP Synthase Assembly

Some described human ATP6 mutations affect only CV function [19,30,54], while others affect both assembly and function [12,23,47,54,55]. To understand the possible effect of *mt-Atp6* mutation on complex V assembly, BN-PAGE followed by WB was performed, using digitonin-solubilized mitochondria from mutant and control cells. The F_1_F_o_ band was assigned by both CV-IGA assays and Western blot (WB). As shown in Figure 5A,B, the complete monomeric F_1_F_o_ fraction was found in similar amount in control and mutant samples. However, a lower molecular weight band with ATPase activity which could correspond to catalytically active F_1_ subcomplexes (CV*, Figure 5A,B), previously reported in human mutants [12,23,47,54,55], was in proportion more intense in the mutant mitochondria. These observations could indicate a slightly more rapid turnover or slower assembly of CV in the mutant cell line. As seen in Figure 2E and Figure 5, the assembly and activity of other OXPHOS complexes was similar in both cell types.

### 2.6. m.8414A>G Mutation Decreases Migration Ability and Tumorigenic Phenotype in TmA6^MUT^ Cells

According to the literature, ATP synthase activity has been associated with proliferation, invasive capacity, and apoptosis resistance in cancer cells [33,38,40]. Indeed, most cancer cells rely on aerobic glycolysis and many of them overexpress the ATPase inhibitor factor 1 (IF1), which affects cancer development through different effects such as cristae shaping or inhibition of ATP hydrolysis, among others [56]. In addition, several mt-ATP6 mutations have been detected in different human cancer cell lines and solid tumors [57,58,59,60], next to the 8414 position and affecting aH5 helix (Figure 6A). All these aspects, together with the higher glycolysis rate, ROS production and sensitivity to DCA (a metabolic drug being tested for cancer treatment in humans) observed in TmA6^MUT^ cells, prompted us to evaluate aspects related to the potential tumorigenic phenotype of this cell line. First, we measured MHC-I surface expression in A6 mutant cells, as some tumor cells lose this factor to escape from immune system surveillance [61]. In this case, TmA6^MUT^ expression of MHC-I is increased two-fold compared to that of control cells (Figure 6B) and it is further increased by DCA treatment (Figure 6D). To test the effect of DCA on mutant cells, we measured cell viability and apoptosis after incubation with either 5 or 25 mM DCA for 72 h. As shown in Figure 6C, cell growth of mutant cells is inhibited 20% and 50% upon 5 or 25 mM DCA treatment, respectively, when compared to non-treated cells. In contrast, control cells’ growth rate is not affected by the treatment (Figure 6C, left panel). However, although DCA does not induce significant differences in cell death in either control or mutant cells at the lowest concentration tested, it promotes apoptosis in both cell lines when used at 25 mM, being significantly higher in complex V mutant cells (Figure 6C right panel). Re-evaluation of mt-ROS levels after incubation of 25 mM DCA for 72 h indicates a three-fold increase in superoxide levels of mutant cells vs the 1.4-fold increase observed in TmControl cells (Figure 6E). Finally, we analyzed the levels of the hypoxia inducible factor 1α (HIF-1α), a transcription factor that regulates the expression of genes implicated in glycolysis and plays a key role in cancer metabolism reprogramming [62], in both control and mutant cells. The analysis was performed by WB (not shown) and immunocytochemistry followed by flow cytometry in basal conditions and after DCA treatment (Figure 6F,G). As shown in Figure 6F, expression of HIF-1α under basal conditions in mutant cells, although statistically non-significant, is slightly lower than that observed in control cells, and DCA treatment did not differentially modify the expression levels of this protein among the two cell lines (Figure 6G).

To further analyse the tumor phenotype of TmA6^MUT^ cells, we evaluated the migration capacity of control and mutant cells using microfluidics technology. For this purpose, cells were 3D-cultured in collagen-based hydrogels within the microfluidic devices and time-lapse images were taken every 20 min for 24 h, ensuring that the tracked cells were embedded within the 3D network. Then, we calculated the average (Vmean: total distance travelled vs time) and effective (Veff: end to end distance travelled vs time) speed of both cell types. As shown in Figure 6H, although there was no difference in Veff between control and mutant cells, Vmean was significantly lower in TmA6^MUT^ cells, indicating lower migration ability.

## 3. Discussion

Mutations affecting mitochondrial complex V structural subunits or assembly factors are the molecular cause of a wide variety of mitochondrial disorders in humans. Especially relevant are those changes located in the mtDNA-encoded *ATP6* gene which concentrates most of the mutations in mitochondrial ATP synthase characterized up to date [30]. One of the main challenges of mitochondrial diseases is the clinical heterogeneity observed in patients which may cause difficulties in diagnosis. The other one is the treatment of patients as, in most of the cases, there is no effective cure for mitochondrial disorders and treatments are focused on managing symptoms [63]. Although there are different cellular models that are very useful for unravelling the molecular mechanisms of mitochondrial diseases due to mutations in mtDNA, animal models are a more suitable option to study these disorders in the context of a complete organism. Since these mutations have not been found in nature in mice, to generate such models it is crucial to obtain mouse cell lines harboring mtDNA mutations.

In this work, we reported the generation and characterization of the first *mt-Atp6* gene pathological mutation in mouse cells, a G to A transition at position 8414 that induces an amino acid change from Asn to Ser in a highly conserved position. The substitution affects an exposed and functional residue of the protein (p.N163S) which is located in aH5 helix and appears to have an impact on the interaction between subunit a, and the c-ring, as well as on proton translocation from the mitochondrial IMS to the matrix. The phenotypic consequences of this mutation reproduce the effects of MT-ATP6 mutations in human diseases: (i) increased dependence on glycolysis demonstrated by a higher glucose consumption and lactate production, and by the increased sensibility to glycolysis inhibitors such as sodium iodoacetate; (ii) defective OXPHOS activity, shown by their incapacity to survive in non-fermentative culture media, by the reduction in their oxygen consumption rate and the higher sensibility to DCA which forces OXPHOS; (iii) ATP synthesis impairment and higher sensitivity to oligomycin; (iv) increased ROS generation and decreased mitochondrial mass; (v) mitochondrial membrane potential increase when forced to use OXPHOS; and (vi) F_1_-subcomplexes accumulation. Thus, this cell line could be of great interest to develop mouse models of human diseases due to mutations in *MT-ATP6* gene.

The relevance of Helix 5 of subunit a in the function of ATP-synthase is evidenced by the high number of pathological mutations that have been found affecting this domain. As aH6, aH5 is intimately associated with the c-ring that constitutes the membrane region of the enzyme’s rotor (Figure 1D right panel) [64]. aH5 is kinked due to the presence of a proline at position 153 that enables the helix to follow the curvature of the c-ring and to seal the two hydrophilic pockets that connect the interface between subunit a and the ring to the IMS and the matrix [30]. In our mutant cell line, a substitution of an Asn by Ser at position 163 is the molecular cause for the altered phenotype observed. Although such substitution does not seem to constitute a very drastic change, as both are non-charged polar amino acids and their replacement only leads to small hydrophobicity changes, it is located in a highly conserved position, flanked by two amino acids that are mutated in a very similar way (p.A162V [43] and p.I164V [42]) in LHON patients. The molecular mechanism by which these mutations induce pathology in humans is not yet known [30] but it could be similar to that of p.N163S. Thus, TmA6^MUT^ cells could be an appropriate model to study disorders due to alterations in this region of mitochondrial ATP synthase subunit a.

As shown in Figure 7A, the p.N163S substitution disrupts an intramolecular H-bond between the side chains of N163 and Q210 of the aH6 that contributes to stabilize and pack both helices in an optimal conformation for defining the proton pathway. Moreover, the lateral chain of Asn 163 faces E59 of the c-ring, contributing to define the subunit a/c-ring interface. In addition, the substitution of an Asn for Ser, an amino acid with a smaller side chain, locally diminishes the solvent exposition of subunit a inducing an increase in the distance with the c-ring interface that can be detrimental for the suitable fitting of both subunits and, in consequence for c-ring rotation (Figure 7A), reducing in this way the synthesis of ATP. On the other hand, this mutation locally diminishes the positive electrostatic surface potential in a highly conserved region that also includes R159, and is essential for preventing leaks of proton by electrostatic repulsion during their transfer from the IMS entry channel to the matrix exit channel (Figure 7B) [65]. Thus, the reduction of positive surface potential may promote a retention of protons in the channel and a defective release to the matrix what could hinder c-ring rotation and therefore, ATP synthesis.

In recent years, complex V functions have been corelated with hallmarks of cancer cells such as proliferation, invasive behavior, and evasion of cell death [33]. Indeed, some carcinomas present MT-ATP6 mutations [34,35] while others downregulate the expression of the catalytic subunit β-F_1_-ATPase [36,37] reducing in that way OXPHOS activity to favor the metabolic switch to a Warburg phenotype that promotes growth and cell survival [68]. In addition, it has been reported that several tumor types such as colon, lung, breast, and ovarian carcinomas upregulate the expression of ATPase inhibitory factor 1 (IF1) [38] which favors cell proliferation by blocking ATP hydrolysis and, consequently, preserving the amount of ATP [38,39]. More recently, different effects of IF1 on cancer development have been proposed [56]. In this sense, IF1 has been reported to have effects on metastasis promotion and cristae shaping [68,69], as well as on inhibition of ATP hydrolysis and escape from apoptosis. In addition, IF1 seems to modulate both ROS levels and ATP synthesis [70,71] during oxidative phosphorylation. In this work, although, our mutant cells show features which are typical of many cancer cells such as a higher sensitivity to DCA treatment and increased ROS levels, we observed that the p.N163S substitution in MT-APT6 reduces several cell features related to the tumorigenic and invasive phenotype of the cells. Thus, (i) the MHC-I expression, contrary to the downregulation observed in many cancer cells, is higher than in controls both in basal conditions and after DCA treatment; (ii) the HIF-1α levels are slightly reduced in basal conditions compared to control cells; and (iii) the migration capacity is diminished, as shown by a decrease in the average speed of mutant cells measured in vivo using microfluidic devices. This apparent reduction of tumorigenic potential and, especially, of migration ability observed in mutant cells could be explained by the decrease of ATP availability due to the severe reduction in its synthesis by complex V, although other causes cannot be ruled out. Our model suggests a protective role of ATP synthesis inhibition against tumor transformation and migration which can support the new therapeutic strategies proposing mitochondrial ATP synthase as a target in the treatment of some types of cancer with an OXPHOS-dependent metabolism [40,41].

## 4. Materials and Methods

### 4.1. Cell Lines and Media

All cell lines were grown in DMEM (GibcoBRL) supplemented with 5% FBS (fetal bovine serum, Gibco BRL). MtDNA-less mouse cells ρ^0^L929^hygro^ were generated by long-term exposure of L929 mouse cell line with high concentrations of Ethidium Bromide (EthBr) and transfection with the hygrocassette-containing plasmid pcDNA3.1 (Invitrogen, Waltham, MA, USA), as previously described [45,72]. Transmitochondrial cells were generated by transference of mitochondria from platelets or enucleated cell lines to mtDNA depleted cells, as described elsewhere [45,72]. Afterwards, they were isolated by growing them in DMEM supplemented with 5% dialyzed FBS and 10 µg/mL of puromycin (SIGMA) or 500 µg/mL of hygromycin (Gibco BRL). BcTMPII.1 cells were derived by random mutagenesis of TmBalb/cJ cells as previously described [44,46] and harbor an A to G transition at position 8414 within the *mt-Atp6* gene.

### 4.2. DNA Analysis

Total DNA from cell lines was extracted using standard procedures. The complete mtDNA was amplified and sequenced as described before [73]. Primers were designed using the reference sequence (NC_005089) [74].

### 4.3. RFLP Analysis

To confirm the presence of the mutation RFLP analysis was achieved. Thus, a 571 bp fragment was amplified by PCR with the following primers: (i) Fw, CCAAAAGGACGAACATGAACCC (positions 8113–8134) and (ii) Rev, AGGAGGGCTGAAAAGGCTCC (positions 8664–8683, reverse primer).

The G8414 mutant version creates a recognition site for NheI and produces two bands of 298 and 273 bp upon digestion with this enzyme. Fragments were analyzed by electrophoresis in 5% polyacrylamide-TBE gel.

### 4.4. Growth Measurements

Growth capacity in galactose containing medium was determined by plating 5 × 10^4^ cells on 12 wells test plates in 2 mL of the appropriate medium (DMEM, which contains 4.5 mg of glucose/mL supplemented with 5% FBS, or DMEM lacking glucose and containing 0.9 mg of galactose/mL, supplemented with 5% dialyzed FBS), incubating them at 37 °C for 5 days. Cell counts were performed at daily intervals [52].

### 4.5. Oxygen Consumption Measurements

O_2_ consumption determinations in intact or in digitonin-permeabilized cells were carried out in an Oxytherm Clark-type electrode (Hansatech), as previously described [75], with small modifications [45].

### 4.6. Glucose and Lactate Measurements

Glucose consumption and lactate production were measured spectrophotometrically in cell culture media collected from cultures at around 80% of confluence 24 h after medium renewal.

Glucose concentration was determined using the glucose oxidase/peroxidase kit from BioSystems. Briefly, 10 µL of diluted cell culture medium was mixed with 1 mL of reagent and absorbance at 500 nm was measured after 15 min of RT incubation. Concentration was calculated using standards of known concentration. For lactate determination, the colorimetric “L-Lactate Assay Kit” from Abcam was used. Samples were prepared following the manufacturer’s instructions and absorbance at 450 nm was measured in a 96-well plate reader.

### 4.7. Cell Growth Assays

The effect of m.8414A>G mtDNA mutation on sodium iodoacetate, oligomycin or DCA cytostatic ability was evaluated using the MTT reduction assay according to Mosmann et al. [76]. Briefly, cells were plated into flat-bottomed 96-well microtiter plates at a density of 5 × 10^3^ cells per ml. Then, cells were cultured for three days to be allowed to reach exponential growth rate before drug additions. Afterwards, cells were cultured in the presence of different inhibitors concentrations for 48 h. After drug exposure, cells were fed with fresh medium and allowed to grow for 2 population doubling times. At the end of the recovery period, plates were incubated with fresh medium and MTT for 4 h in a humidified atmosphere at 37 °C, formazan crystals were solubilized with DMSO and OD at 550 nm was read. The results obtained were given as relative values to the untreated control in percentage, and inhibitory concentration 50 (IC_50_) was determined as the drug concentration required to reduce the absorbance to half that of the control. All experiments were performed at least in triplicate.

### 4.8. ATP Synthesis Measurements

ATP synthesis driven by the different substrates was measured in digitonin-permeabilized cells (2 × 10^6^ cells) using a kinetic luminescence assay, as previously described [77].

### 4.9. Flow Cytometry Analysis

Mitochondrial superoxide production, mitochondrial mass, mitochondrial membrane potential (MMP), MHC-I surface expression and cell death were measured in basal conditions and after DCA treatment by flow cytometry after cell staining with specific fluorescent probes, using a FACScalibur flow cytometer (BD Biosciences).

For mitochondrial superoxide production, cells were incubated with MitoSOX™ (5 µM, ThermoFisher, Waltham, MA, USA) for 30 min at 37 °C.

For mitochondrial mass, cells were stained with MitoTracker Green FM (200 nM, ThermoFisher), a mitochondrial dye that stains mitochondria regardless of Ψm, following the manufacturer’s recommendations.

For differences in Ψm due to the presence of the m.8414A>G mutation or to DCA treatment, cells were incubated with DiOC_6_ at 20 nM for 30 min or TMRE at 60 nM for 15 min at 37 °C prior to flow cytometry analysis. When indicated, the uncoupler FCCP (10 µM) was added and incubated together with the fluorescent probe.

For MHC-I surface expression, cells were incubated with an anti-mouse MHC-I antibody (FITC Mouse Anti-Mouse H-2Kk, BD Biosciences, East Rutherford, NJ, USA), in PBS with 5% FBS, for 30 min at 4 °C. Next, cells were washed and, its expression was determined by flow cytometry.

Finally, cell death induced by DCA treatment was analyzed after incubation with annexin-V-FITC, which binds to the phosphatidylserine exposed in the cell surface, and 7-AAD (7-aminoactinomycin) which binds DNA, indicating cell damage, in annexin binding buffer (140 mM NaCl, 2.5 mM CaCl_2_, 10 mM HEPES/NaOH pH 7.4) for 15 min at room temperature.

All data from the cytometer were analyzed using the FlowJo software.

### 4.10. mtDNA Copy Number Measurement

mtDNA copy number was measured by qPCR with a LightCycler^®^ 2.0 Instrument (Roche, Basel, Switzerland) and LightCycler Fast-Start DNA MasterPLUS SYBR Green I (Roche) in a 20 µL final volume, as recommended by the manufacturer. Total cellular DNA was used as a template and was amplified with specific oligodeoxynucleotide for mt-Co2 (NC_005089) and Sdha (AK049441) which was used as a reference for nuclear DNA content, as previously described [52].

### 4.11. Blue Native Polyacrylamide Electrophoresis

Mitochondria were isolated from cultured cell lines according to Schägger (1995) [78], with some modifications [79]. Digitonin-solubilized mitochondrial proteins (100 µg) were separated on blue native gradient gels (3–13% acrylamide, Invitrogen^TM^, Waltham, MA, USA).

### 4.12. Respiratory Complex and Supercomplex Analysis

After electrophoresis, gels were further processed for either western blotting or In Gel Activity IGA assays.

In order to analyze the assembly status of respiratory complexes and supercomplexes, proteins separated by BN were transferred onto PVDF membranes and probed with specific antibodies that recognize the different respiratory complexes (anti-NDUFA9, anti-SDHA, anti-Uqcrc1 and anti-COI for complexes I, II, III, IV respectively, all of them from Invitrogen, and anti-β-F_1_-ATPase, for complex V, kindly gifted by Dr. J.M. Cuezva). Then, they were incubated with their corresponding secondary antibody, tagged with peroxidase (ThermoFisher Scientific, Waltham, MA, USA). Finally, protein levels were evaluated by immunoblot, using an AmershamTM Imager 600 to document the signal.

Complex I IGA assays were performed after BN-PAGE by incubating the gels at room temperature in the presence of 0.1 M Tris-HCl pH 7.4, 0.14 mM NADH and 1 mg/mL of nitro-blue tetrazolium (NBT) as previously described [80]. Complex V IGA staining was performed by incubating the gels in 34 mM Tris, 270 mM glycine, 14 mM MgSO_4_, 0.2% Pb(NO_3_)_2_ and 8 mM ATP at pH 7.8, as described before [81]. After incubation, the gels were photographed with a digital camera using as background a white translucid screen for Complex I and Complex IV and a black screen for Complex V [81].

### 4.13. HIF-1α Expression Evaluation

HIF-1α expression was evaluated by flow cytometry using a specific antibody anti- HIF-1α. 10^5^ cells were seeded in a 48-well-plate and let them attach for 24 h at 37 °C and 5% CO_2_. Next day, cell culture media was removed, cells were trypsinized, transferred into a flow cytometry tube and washed with PBS three times by centrifugation. Then, cells were fixed using 4% PFA for 10 min at 4 °C. After performing three washes with PBS, cells were permeabilized with 0.1% saponin and incubated for 20 min at RT. Subsequently, saponin was removed by washing three more times and primary antibody anti- HIF-1α (1:700, Novus Bio #NB100-449) was incubated for 1h at RT. Secondary anti-rabbit antibody Alexa Fluor^®^ 488 (1:1000, Invitrogen #A11034) was incubated for 10 min at RT after removing primary antibody and washing three times. Finally, fluorescence intensity was assessed by flow cytometry. To avoid unspecific interactions, control cells were only incubated with the secondary labeled-antibody and compared fluorescence intensities between cells stained with both primary and secondary antibodies. Data obtained was analyzed using FlowJo software.

### 4.14. Cell Migration Analysis

The ability of cells to migrate was evaluated using microfluidic devices. Microdevices were fabricated in PDMS following the methodology described by Shin et al. [82]. TmControl and TmA6^MUT^ cells were cultured within the microfluidic chip embedded in 3D-collagen hydrogels and migration was evaluated by time-lapse imaging carried out with a Nikon D-Eclipse Ti Microscope with a 10X objective, acquiring phase contrast images every 20 min for 24 h as described before [83]. Briefly, cells were mixed with collagen type I hydrogel (2.5 mg/mL BD Biosciences) to a final concentration of 2 × 10^5^ cells/mL. Then, the hydrogel dilution was pipetted into the device’s central gel chamber and confined by surface tension. Once in place, collagen gel solution was polymerized in a humidity chamber at 37 °C and 5% CO_2_ for 20 min. After that, the gel was hydrated with culture medium and incubated overnight for matrix stabilization and cell adhesion. Time-lapse imaging was obtained as described above, at 37 °C, 5% CO_2_ and 95% of humidity. Cell tracking was performed using a hand coded semi-automatic MATLAB script described in previous studies [83,84]. Acquired data were then analyzed in terms of mean and effective speeds. By comparing pixel intensities and using matrix convolution techniques, the software was able to find and track cell centroids, request visual correction from the user, and post-process the migration results. These trajectories were used to extract the cell mean (Vmean) and effective (Veff) velocities. Vmean and Veff definitions are shown in Figure 5H.

### 4.15. Bioinformatic Tools

The structural model of MT-ATP6 from mouse was obtained from AlphaFold protein structure database [66,67] and used to create N163S mutant using mutagenesis tool embedded in PyMOL Molecular Graphics System [85] Alpha fold models conservation scores were calculated by the ConSurf server [86] using solely the protein 513 sequence of interest as input. ESP for WT and N163 variant was calculated using the APBS-PDB2PQR software suite (https://www.poissonboltzmann.org/ (accessed on 17 November 2022)) at pH 7.4 and 150 mM salt concentration. The PyMOL Molecular Graphics System was used to produce the structural figures.

### 4.16. Statistical Analysis

Results, displayed as mean ± SEM, were statistically analyzed by Fisher’s PLSD post hoc test from ANOVA. Data were analyzed with the statistical package StatView 5.0 (SAA Institute, Mesa, AZ, USA) or with GraphPad Prism 8.1.2 software. In all cases, differences were considered statistically significant at *p* ≤ 0.05. The significance of the observed differences in migration assays, was obtained via Welch’s test of Vmean and Mann–Whitney test of Veff.

## Figures and Tables

**Figure 1 ijms-24-01300-f001:**
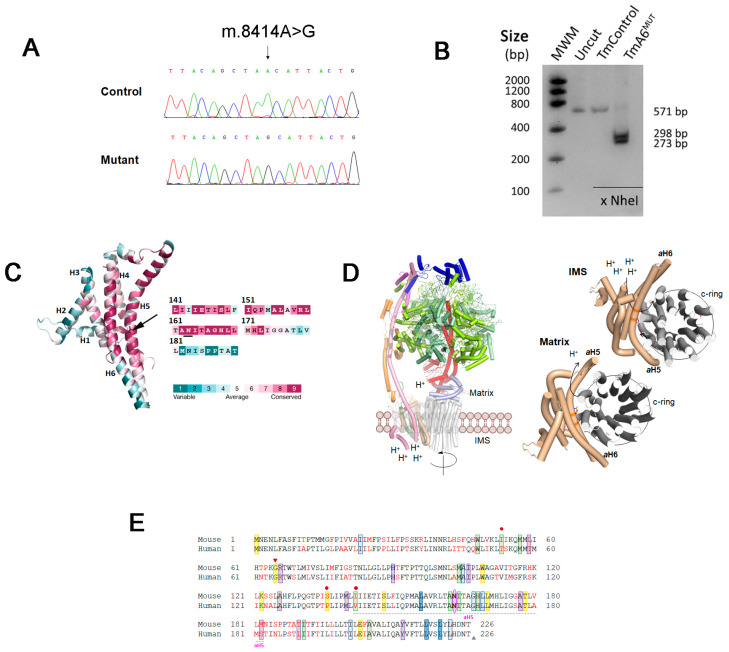
Analysis of m.8414A>G mutation in mt-Atp6 gene and protein: (**A**) chromatogram showing the m.8414A>G mutation within the *mt-Atp6* gene in mutant BcTMP11.1 cells; (**B**) RFLP analysis of the mutation in transmitochondrial mutant (TmA6^MUT^) and control (TmControl) cells. The presence of the mutation creates a recognition site for NheI; (**C**) ConSurf analysis of F_o_F_1_ ATP synthase subunit a (MT-ATP6) from *Mus musculus* shown as cartoon colored according to the conservation score calculated by ConSurf server (left). The protein is folded in six transmembrane α-helixes connected by loops. The N163 (represented with sticks, black arrow) is an exposed residue located in a highly conserved region of helix aH5. As shown in the aH5 sequence (right), N163 is a highly conserved amino acid; (**D**) structure of mitochondrial ATP synthase (left) and interaction of subunit a with the c-ring (right). Left: Cartoon representation of subunit composition of the bovine ATP synthase monomer (PDB 6zpo). The catalytic subunits α and β (upper part) are lemon yellow and pale green, respectively. The asymmetrical central stalk subunits ƴ, δ and ε are red, light blue and violet, respectively. The peripheral stalk subunits OSCP, F6, b and d are blue, violet, pink and orange, respectively. The membrane domain composed of the c-ring in contact with subunit a, and the supernumerary subunit A6L are gray, wheat and green forest, respectively. To simplify the representation, the rest of supernumerary subunits e, f, g, j and k have been omitted. Right: View of the entire c-ring and subunit a from the intermembrane space and matrix (upper and bottom panels, respectively) and the path of protons, showing subunit a and c-ring (PDB 6zpo). The side chains of the essential residues for proton transfer, E59 and R159 (c-ring and subunit a, respectively), and the mutated N163 residue are shown as CPK colored sticks with C atoms in grey for c-ring, and in wheat for the subunit a; and (**E**) alignment of human and murine MT-ATP6 protein sequences. Reported human pathological mutations and the mouse mutation (pink) are highlighted in the sequence with the following color code: green for LOHN causing mutations, blue for NARP, MILS or Leigh syndromes (dark blue: frequent mutations; light blue: rare mutations), purple for autism or schizophrenia-related mutations and yellow for “other mutations”. Triangles indicate mutations creating a stop codon (red: insertion; grey: deletion). Red dots indicate the presence of mutations in the human protein that introduce an amino acid change identical to that of the murine protein.

**Figure 2 ijms-24-01300-f002:**
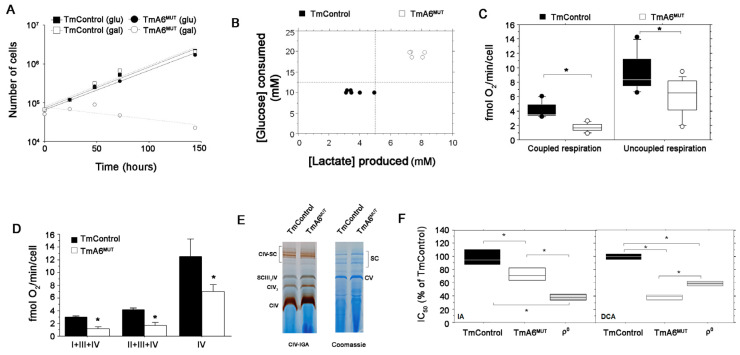
Analysis of mutant cells glucose dependence and OXPHOS performance: (**A**) representative growing curve of TmControl and TmA6^MUT^ in two different culture media: Glucose containing medium and galactose containing medium in which only OXPHOS competent cells are able to grow; (**B**) glucose consumption and lactate production by TmControl and TmA6^MUT^ cell semi-confluent cultures in 24 h. The graph represents the concentrations relationship between lactate produced and glucose consumed, measured in cell culture media collected from a 24 h culture (80% of confluence); (**C**) endogenous (left) and maximal (right) oxygen consumption rate measurement in intact cells (*n* = 9 and 13 for TmControl and TmA6^MUT^, respectively; *p* < 0.0001 for endogenous respiration and *p* < 0.05 for maximal respiration); (**D**) oxygen consumption of digitonin-permeabilized cells in the presence of specific electron donors for complex I (glutamate + malate), complex II + III (succinate + G3P) and complex IV (TMPD) (*n*= 2 and 3 for TmControl and TmA6^MUT^, respectively; *p* = 0.0071, *p* = 0.0072 and *p* = 0.048 for complex I + III + IV, complex II + III + IV and complex IV, respectively). (**E**) Pattern of mitochondrial complexes and supercomplexes analysed by BN-PAGE followed by enzymatic In Gel Activity for complex IV (left) or by coomassie staining (right); and (**F**) IC_50_ expressed as % of that of TmControl cells for the indicated inhibitors in TmA6^MUT^ and ρ^0^ cells (*n* = 3 in all cases for IA and *n* = 3, 3 and 2 for TmControl, TmA6^MUT^ and ρ^0^ cells, respectively, for DCA; *p* < 0.05 for all the pairs and inhibitors). All values are given as mean ± SD of the mean. Asterisks indicate significant differences respect to control cells, tested by ANOVA post-hoc Fisher PLSD (*p* < 0.05 or as indicated in each panel description).

**Figure 3 ijms-24-01300-f003:**
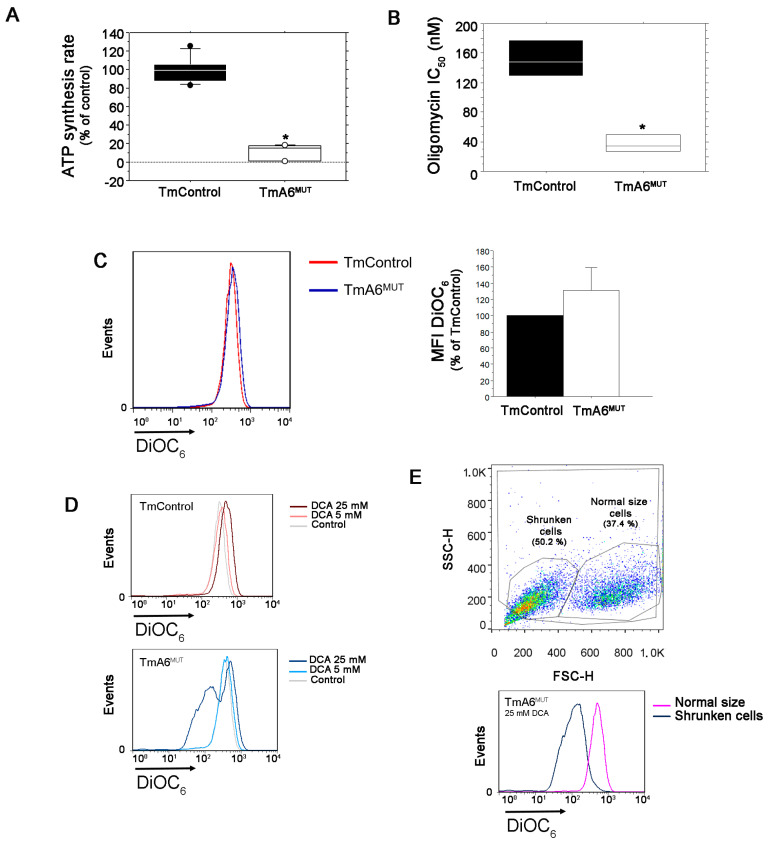
Analysis of mutant cells mitochondrial ATP synthesis activity and membrane potential: (**A**) ATP synthesis rate measurement in control and mutant cells (*n* = 9 and 6 for control and mutant cells, respectively; *p* < 0.0001); (**B**) IC_50_ for olygomycin in TmControl and TmA6^MUT^ cells (*n* = 4 and *p* = 0.0008); (**C**) analysis of mitochondrial membrane potential (ΔΨm) in control and mutant cells. Changes in ΔΨm were determined by staining with DiOC6 and analyzed by flow cytometry. As indicated in the legend, red histogram corresponds to TmControl cells, while blue histogram corresponds to mutant cells. The left panel shows a representative histogram of ΔΨm measurement in both cell lines under basal conditions (the further to the right the peak is, the higher the membrane potential) whereas the right panel represents the average ΔΨm of mutant cells relative to controls from two independent experiments; (**D**) ΔΨm evaluation in both cell lines after treatment with different concentrations of DCA; and (**E**) Mutant cells were further separated into different populations, depending on their size (normal size and shrunken, left), and their corresponding ΔΨm was compared (right). All values are given as mean ± SD of the mean. Asterisks indicate significant differences respect to control cells, tested by ANOVA post-hoc Fisher PLSD (*p* < 0.05 or as indicated).

**Figure 4 ijms-24-01300-f004:**
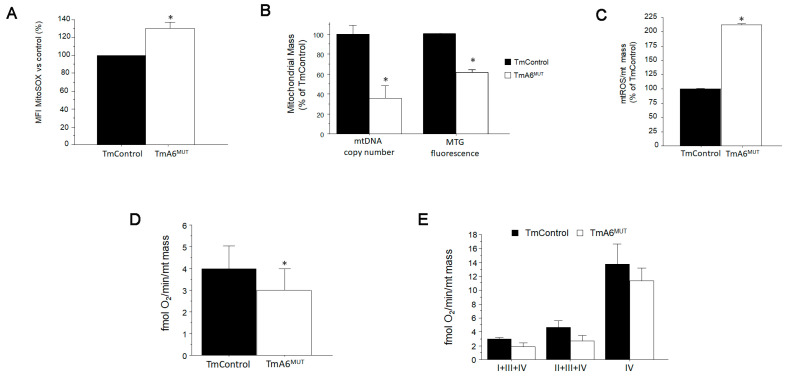
Analysis of mitochondrial ROS (mtROS) production and mitochondrial mass: (**A**) mtROS production in control and mutant cells was analyzed by flow cytometry after staining with MitoSOX^TM^ (*n* = 2 for both cell lines, *p* = 0.0208); (**B**) mitochondrial mass was estimated by two different ways: mtDNA copy number quantification by qPCR (left) and flow cytometry (right) after mitochondria staining with MitoTracker Green^TM^, a fluorescent probe that specifically stains mitochondria regardless of ΔΨm (*n* = 5 for mtDNA copy number quantification in both cell lines, *p* < 0.001 and *n* = 2 for mitochondrial mass estimation using MitoTracker Green; *p* = 0.0002); (**C**) mtROS production per mitochondria. Estimation of mitochondrial superoxide production normalized to mitochondrial mass (*n* = 3 in both cases, *p* = 0.025); (**D**) oxygen consumption (coupled respiration) normalized per mitochondrial mass (n = 9 and 12 for control and mutant respectively, *p* = 0.046); and (**E**) oxygen consumption of digitonin-permeabilized cells with specific substrates normalized per mitochondrial mass (*n* = 2 and 3 for control and mutant respectively and *p* > 0.05 in all cases). In all cases, the normalization by mitochondrial mass was done using the data from MitoTracker Green^TM^ staining. All values are expressed as mean ± SD of the mean. Asterisks indicate significant differences respect to control cells, tested by ANOVA post-hoc Fisher PLSD (*p* < 0.05 or as indicated).

**Figure 5 ijms-24-01300-f005:**
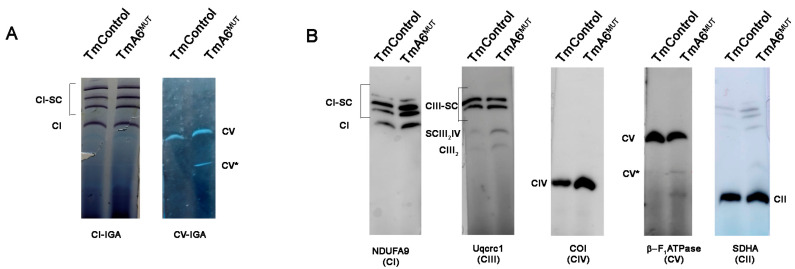
OXPHOS complex and supercomplex assembly analysis: (**A**) pattern of mitochondrial complexes and supercomplexes containing CI (left panel) and CV (right panel) analyzed by BN-PAGE followed by enzymatic IGA of the indicated complexes; and (**B**) immunodetection of the different OXPHOS complexes and supercomplexes in TmControl and TmA6^MUT^ mitochondria. Antibodies: anti-NDUFA9 for complex I, anti-Uqcrc1 for complex III, anti-Cox1 for complex IV, anti-β-F_1_ ATPase for complex V and anti-SDHA for complex II. Notice the higher presence of a band corresponding to CV subcomplexes (CV*) and of low molecular weight antibody-reacting material in mutant mitochondria.

**Figure 6 ijms-24-01300-f006:**
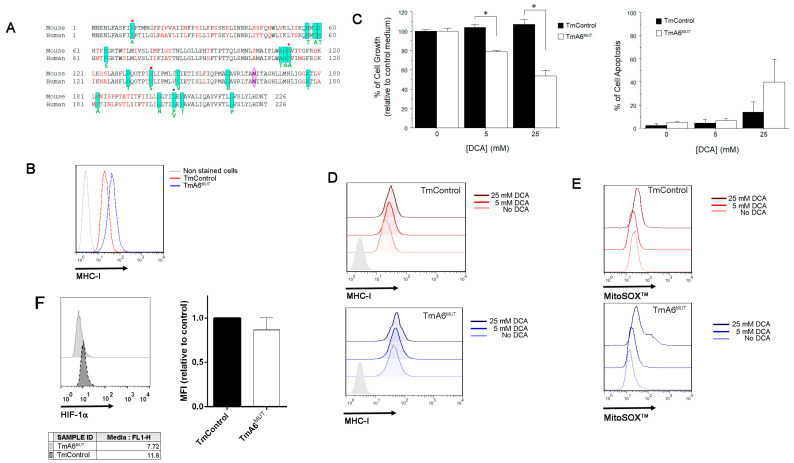
Tumor phenotype analysis: (**A**) MT-ATP6 mutations found in human cancers are highlighted in green and the corresponding amino acid change is indicated under the human sequence [33]. Red dots indicate that the mutation promotes a change that is identical to the murine sequence. Blue dot indicates two different changes in the same position; (**B**) MHC-I expression analysis in untreated TmControl and TmA6^MUT^ cells determined by flow cytometry; (**C**) Cell growth (left) and cell death (right) after treatment with different DCA concentrations. Cell growth effect was measured using the MTT reduction method (*n* = 2 for each cell line and treatment; *p* = 0.0109 and 0.0108 for 5 mM and 25 mM DCA, respectively). To evaluate cell death, cells were simultaneously stained with annexin-V-FITC and 7 AAD and fluorescence was measured by flow cytometry (*n*= 3 for each cell line and treatment); (**D**,**E**) effect of different DCA concentrations on MHC-I expression (**D**) and on mitochondrial superoxide production (**E**) in TmControl (upper panels) and TmA6^MUT^ (lower panels) cell lines by flow cytometry; (**F**,**G**) analysis of HIF-1α expression levels in basal conditions; (**F**) and after treatment with different concentrations of DCA; (**G**) by immunocytochemistry followed by flow cytometry. CoCl_2_ (500 µM) was included as a control of hypoxia induction effect; and (**H**) Evaluation of cell migration using microfluidic devices. Definition (left panels) and measurement (right panels) of effective (Veff) and mean (Vmean) speed for TmControl and TmA6^MUT^ cells (*n* = 3 independent experiments with 3 technical replicas each; *p* < 0.001). Asterisks indicate statistically significant differences between control and mutant cells: * indicates *p* < 0.05 and *** indicates *p* < 0.001.

**Figure 7 ijms-24-01300-f007:**
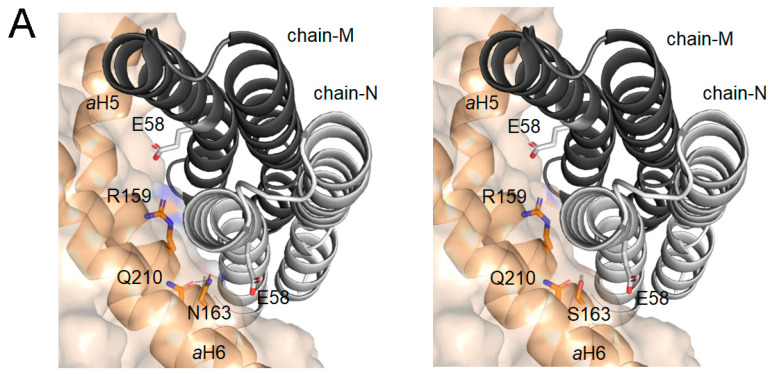
The impact of N163S mutation in the structural model of the mouse ATP synthase subunit a: (**A**) detail of the interface view between the c-ring and a subunit for WT (left panel) and p.N163S (right panel). Chain-N and chain-M of c-ring are displayed with cartoon representation (light and dark gray, respectively). Subunit a is displayed as surface with cartoon representation of the secondary structure elements, helices aH5 and aH6, implicated in the interaction. Relevant residues are represented as sticks and CPK colored; and (**B**) impact of p.N163S mutation on the electrostatic surface potential (ESP) of mouse subunit a. ESP for wild-type (left panel)) and its N163S (right panel) variant was calculated at pH 7.4 and 150 mM of salt using the APBS-PDB2PQR software suite (https://www.poissonboltzmann.org/ (accessed on 17 November 2022)) and then plotted using PyMOL. Position of N163S mutation is indicated by a black arrow. The range of change potential is −5 (red) to +5 (blue). The structural model of mouse subunit a (AF-P00848) was obtained from AlphaFold protein structure database [66,67].

## Data Availability

The data presented in this study are available in Appendix A.

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
