# Peer review of "A Mutation in Mouse MT-ATP6 Gene Induces Respiration Defects and Opposed Effects on the Cell Tumorigenic Phenotype"

_ijms, 2023, doi:10.3390/ijms24021300_

Round 1

Reviewer 1 Report

In this work, the authors reported the generation and characterization of the first mt-Atp6 gene pathological mutation in mouse cells. This is a G - A transition at position 8414 that induces a change from Asn to Ser in a highly conserved position. The authors claim that this mutation reproduces the effects of MT-ATP6 mutations in human diseases that means: increased dependence on glycolysis; defective OXPHOS activity, impairment of ATP synthesis, increased ROS production, decreased mitochondrial mass, increased mitochondrial membrane potential and F1-subcomplexes accumulation. Thus, the authors propose this cell line as mouse model of human diseases due to mutations in MT-ATP6 gene.

In my opinion several aspects in this work should be better analyzed and in this form it is too preliminary.

Major points:

Fig.2 panel B; the oxygen consumption rate is reduced at any entry point of electrons (complex I, III and IV). The authors said that the whole electron transport chain is affected. This is a measurement of oxygen consumption rate of whole cells and not of mitochondrial proteins. In my opinion this can mean that the mitochondrial content is reduced in mutant cells or that, less likely, the mutant cells have a reduction of complex IV activity. In order to evaluate the efficiency of mitochondrial respiratory chain the respiration should be performed on isolated mitochondria or normalized on mitochondrial mass. Also the actvity of mitochondrial respiratory chain complexes shoul be evaluated. 

Fig. 2 panel c; the authors should better describe the parameter evaluated in this panel. The IC50 of mutant cells evaluated in the presence of IA (glycolysis inhibitor) appears to be not significant with respect to control cells (p=0.092) but the authors claimed a reduction of vitality because mutant cells are more glycolytic. As for IA (even if it is not significant), the significant inhibition of IC50 by DCA, that forces mitochondria, is interpreted as dependence on glycolysis. On the contrary, the significant inhibition of IC50 by oligomycin, is not interpreted as dependence on aerobic metabolism but due to mitochondrial OXPHOS damage in mutant cells. The interpretation of these results is confused and should be better explained.

Fig.2 panel D; in my opinion, to elucidate the dependence of mutant cells on glycolysis or mitochondrial OXHPOS respect to control cells, ATP production should be calculated in the presence of oligomycin and/or IA, respectively.

Fig. 2 panel F; the quality of the image is very poor and it is difficult to evaluate. In any case, the authors said that the mitochondrial hyperpolarization in the mutant cells (observed when OXPHOS is forced) is due to abnormal proton flux through ATP Fo portion. To demonstrate this,  I think that mitochondrial membrane potential should be measured spectrofluorimetrically by using safranine in the presence of oligomycin and, in addition, should be evaluated the drop of membrane potential by adding ADP.

Fig 3 B, as for Ros production, the oxygen consumption rate (fig 2 ) should be normalized on mitochondrial mass.

Fig. 4. The authors speculate that F1 subcomplexes (CV*, Fig 4A and B), was in proportion more intense in the mutant mitochondria. It seems that all also F1Fo is more intense as well as all complexes of OXPHOS. It is need to quantify the bands and also normalized by immunoblotting against a mitochondrial membrane protein like a subunit of Tom complex.   

Author Response

Please see the attachment "Answer to revieres RM-L et al."

Reviewer 2 Report

Regretfully, almost all researchers, who work with cultured cells, forget that ALL aminoglycoside antibiotics are mitotoxic. Isolated mitochondria from the cells grown up in the presence of these antibiotics have no respiration. Measurements of oxygen consumption by permeabilized cells do not necessarily indicate mitochondrial respiration. More likely, peroxisomes consume oxygen, particularly in cells with nonrespiring mitochondria. Nevertheless, your experimental work partly overcomes these restrictions. Although I think that in the future, you must consider the effects of aminoglycoside antibiotics.

Round 2

Reviewer 1 Report

The manuscript has been improved.